# Helicity dependent photocurrent in electrically gated $(Bi_{1-x}Sb_x)_2Te_3$ thin films

Yu Pan[1], Qing-Ze Wang[1], Andrew L. Yeats[2], Timothy Pillsbury[1], Thomas C. Flanagan[1], Anthony Richardella[1], Haijun Zhang[3], David D. Awschalom[2], Chao-Xing Liu[1] & Nitin Samarth[1]

Circularly polarized photons are known to generate a directional helicity-dependent photocurrent in three-dimensional topological insulators at room temperature. Surprisingly, the phenomenon is readily observed at photon energies that excite electrons to states far above the spin-momentum locked Dirac cone and the underlying mechanism for the helicity-dependent photocurrent is still not understood. Here we show a comprehensive study of the helicity-dependent photocurrent in $(Bi_{1-x}Sb_x)_2Te_3$ thin films as a function of the incidence angle of the optical excitation, its wavelength and the gate-tuned chemical potential. Our observations allow us to unambiguously identify the circular photo-galvanic effect as the dominant mechanism for the helicity-dependent photocurrent. Additionally, we use an analytical model to relate the directional nature of the photocurrent to asymmetric optical transitions between the topological surface states and bulk bands. The insights we obtain are important for engineering opto-spintronic devices that rely on optical steering of spin and charge currents.

[1] Department of Physics and Materials Research Institute, The Pennsylvania State University, University Park, PA 16802-6300, USA. [2] Institute for Molecular Engineering, University of Chicago, Chicago, IL 60637, USA. [3] National Laboratory of Solid State Microstructures, School of Physics and Collaborative Innovation Center of Advanced Microstructures, Nanjing University, Nanjing 210093, China. Correspondence and requests for materials should be addressed to N.S. (email: nxs16@psu.edu)

Narrow band gap semiconductors such as the Bi-chalcogenides have attracted much contemporary interest as three-dimensional (3D) topological insulators (TIs) that host gapless spin-textured surface states that reside within the bulk band gap[1–7]. The spin-momentum locking in these helical Dirac states provides a unique opportunity for topological spintronics devices that function at technologically relevant temperatures (300 K and above)[8–10]. Optical methods have also been adopted to control electron spin and charge currents in 3D TIs[11–14]. In particular, experiments have shown that circularly polarized light induces a directional helicity-dependent photocurrent (HDPC) in 3D TIs: in other words, light of opposite circular polarization yields a photocurrent propagating in opposite directions[15]. The ready observation of this phenomenon at 300 K promises interesting opportunities for developing opto-spintronic devices wherein electron currents might be steered optically. Progress toward such technological applications is however impeded by a lack of understanding about the physics underlying this phenomenon.

The natural impulse is to immediately attribute the HDPC in 3D TIs to the helical spin texture of the Dirac surface states[15]: circularly polarized photons couple to the spin-momentum-locked topological surface states, yielding a circular photo-galvanic effect (CPGE), a phenomenon well-established in semiconductor quantum wells, where the inversion symmetry breaking is the cause[16, 17]. Time-resolved measurements of the HDPC in 3D TIs seem to confirm the surface-related origin for the

HDPC by showing that the group velocity of the induced photocurrent matches that expected for Dirac surface electrons[18]. However, this interpretation is difficult to reconcile with the fact that the photon energy used in the experiments is about 5 times larger than the bulk energy gap. This means that the optical transitions involved in creating photo-electrons cannot involve the Dirac surface states alone.

In principle, HDPC can also be generated by the photon drag effect, which involves the transfer of both spin angular momentum and translational momentum from circularly polarized photons to electrons[19–21]. It is thus relevant to develop an experimental test that distinguishes between CPGE and photon drag as the origin of the HDPC. Another relevant question is the underlying microscopic mechanism for the HDPC. While both the CPGE and the photon drag effect can be related to optical transitions involving the lowest energy surface states alone, there are several other possibilities. These include the asymmetric scattering of electrons[22], a surface shift current[23], a photocurrent arising from Rashba states[24–26], and a photocurrent arising from excitation into the second excited band of surface states[18, 27]. So far, there have been very few attempts to clearly rule out any of these alternative interpretations.

Here, we describe a comprehensive experimental study of the HDPC in thin films of an electrically gated 3D TI (($Bi_{1-x}Sb_x$)$_2$Te$_3$). By revealing three prominent features in the behavior of the HDPC, we unambiguously identify its physical and microscopic origin. First, we find that the HDPC varies as

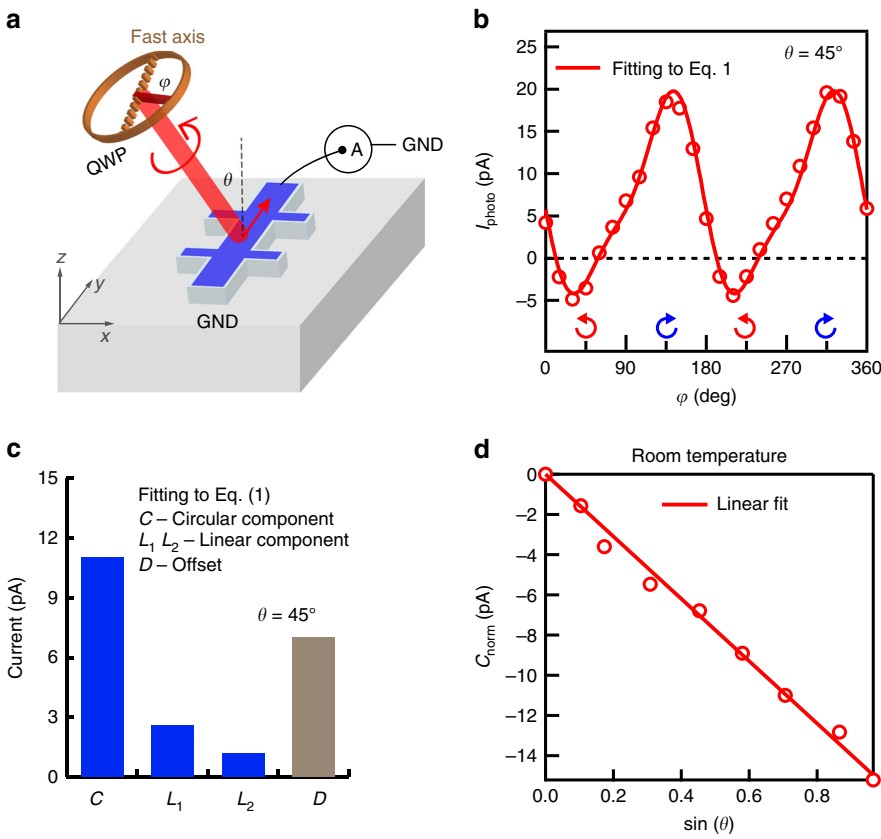

**Fig. 1** Polarization-dependent photocurrent at room temperature. **a** Schematic of the excitation and measurement of photocurrent with obliquely incident excitation (incidence angle $\theta$) in the $x$-$z$ plane. A quarter wave-plate (QWP) is used to tune the polarization of light (by tuning the angle $\varphi$ between the fast axis and the linear polarization of light) and the photocurrent is measured along the $y$-axis. **b** Room temperature photocurrent along the $y$-axis in device A. At $\varphi = 45°$, the laser is left circularly polarized and the photocurrent is negative. At $\varphi = 135°$, the laser is right circularly polarized and the photocurrent is positive. The solid line is a fit to Eq. (1). **c** The coefficients $C$, $L_1$, $L_2$ and $D$ as extracted from the fit in **b**. The amplitude $C$ of the HDPC dominates the polarization-dependent photocurrent. **d** Photocurrent measured at different incidence angles $\theta$. The normalized amplitude $C_{norm}$ of the HDPC plotted as a function of $\sin(\theta)$ shows a linear dependence

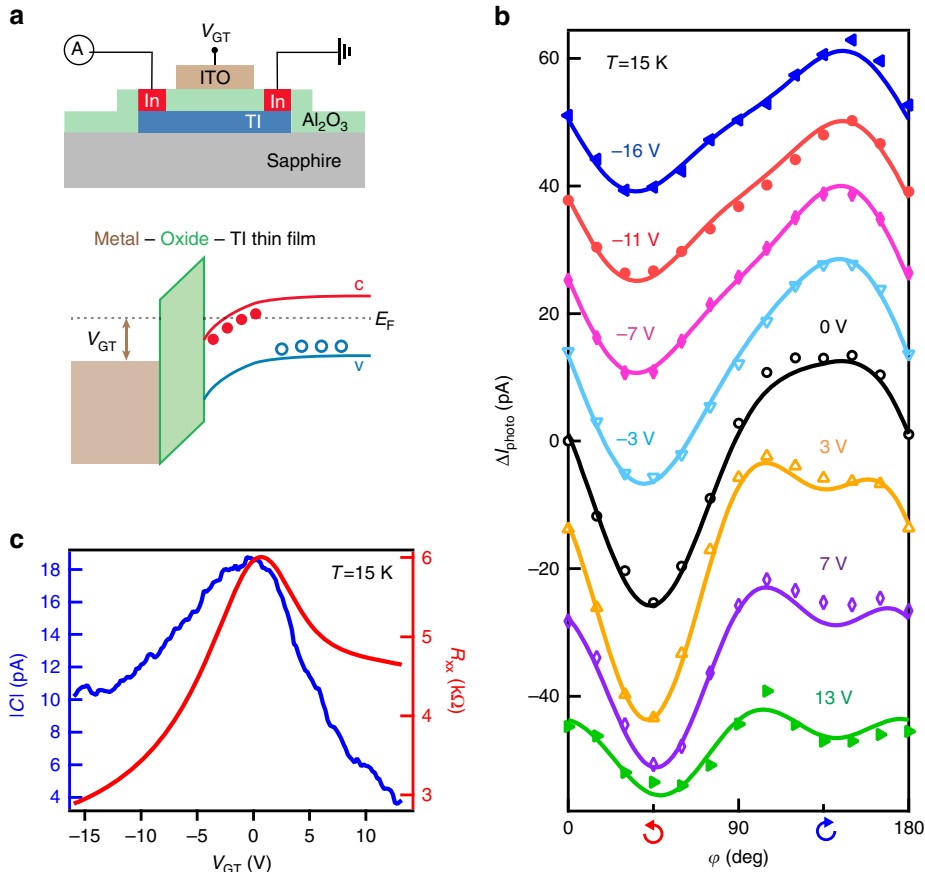

**Fig. 2** Chemical potential dependence of photocurrent in top-gated device B. **a** Sketch of the metal-oxide-TI heterostructure for the top gate. The $Al_2O_3$ is the dielectric layer and the ITO on top serves as a transparent contact for applying the top gate voltage $V_{GT}$. The bottom figure shows the band bending and associated chemical potential moving when a positive $V_{GT}$ is added. **b** The polarization-dependent photocurrent at different $V_{GT}$ is measured at 15 K in device B. The photocurrents are offset for a better display. We fit the polarization-dependent photocurrent at each $V_{GT}$ to Eq. (1), denoted by the *solid line*. **c** Four-probe measurement of the longitudinal resistance $R_{xx}$ as a function of the gate voltage, denoted by the *red curve*. Around zero gate voltage, $R_{xx}$ reaches a maximum, indicating the chemical potential approaches the Dirac point. The absolute value $|C|$ of the amplitude of the HDPC is denoted by the *blue curve*. $|C|$ also reaches a peak around zero gate voltage

$\sin\theta$, where $\theta$ is the angle of incidence of the excitation. Second, by tuning the carrier density with a top gate, we find that the HDPC is a non-monotonic function of the chemical potential and has a peak magnitude when the chemical potential is at the Dirac point of topological surface states. Third, we find that the HDPC varies dramatically with photon energy, reversing sign at a photon energy of $E_p = 1.5$ eV. We show that all these experimental observations are consistent with the CPGE. By combining first principles calculations with an analytical model, we develop a rigorous theoretical explanation that attributes the HDPC to asymmetric optical transitions between topological surface states and bulk states.

## Results

**CPGE as a physical mechanism of the HDPC.** The samples used in our study are 10 or 20 quintuple-layer (QL) thick $(Bi_{1-x}Sb_x)_2Te_3$ thin films grown on sapphire substrates by molecular beam epitaxy. Angle- and polarization-resolved photocurrent measurements are carried out in the geometry shown in Fig. 1a with details described in Methods. Excitation is provided by a 633 nm laser incident in the $x$-$z$ plane with an incidence angle of $\theta$. We measure the photocurrent along the $y$-axis as a function of the angle $\varphi$ between the fast axis of the quarter-wave plate and the initial linear polarization of the laser. The photocurrent along the $x$-axis is independent of the helicity of the light

(Supplementary Fig. 1). As shown in Fig. 1b for the incident angle $\theta = 45°$, the photocurrent in a 20 QL $(Bi_{0.52}Sb_{0.48})_2Te_3$ thin film—device A—at room temperature oscillates when the light polarization is tuned from linearly polarized ($\varphi = 0°$, 90°, 180°) to circularly polarized ($\varphi = 45°$, 135°). Note that the photocurrent reverses sign as the helicity of the light reverses from left-circular polarized ($\varphi = 45°$) to right-circular polarized ($\varphi = 135°$). To extract the origin of the photocurrent, we fit the $\varphi$ dependence of the photocurrent with the expression

$$I(\varphi) = C\sin(2\varphi) + L_1\sin(4\varphi) + L_2\cos(4\varphi) + D, \quad (1)$$

where the first term $C\sin(2\varphi)$ describes the contribution from circularly polarized light (the HDPC contribution), the second and third terms, $L_1\sin(4\varphi)$ and $L_2\cos(4\varphi)$, denote the contributions from linearly polarized light and the last term $D$ identifies an offset photocurrent independent of the polarization. The fit to this equation (*solid line* in Fig. 1b) agrees well with the experimental data. We also obtain the magnitudes of $C$, $L_1$, $L_2$ and $D$ from the fits (Fig. 1c). The offset $D$ is mainly contributed by a dark current, which we cannot eliminate without laser. The linear polarization-dependent photocurrents $L_1\sin(4\varphi)$ and $L_2\cos(4\varphi)$ are not major components in the photocurrent we observe. Several experiments focusing on the linear polarization-generated photocurrent in TIs found its origin from the photon

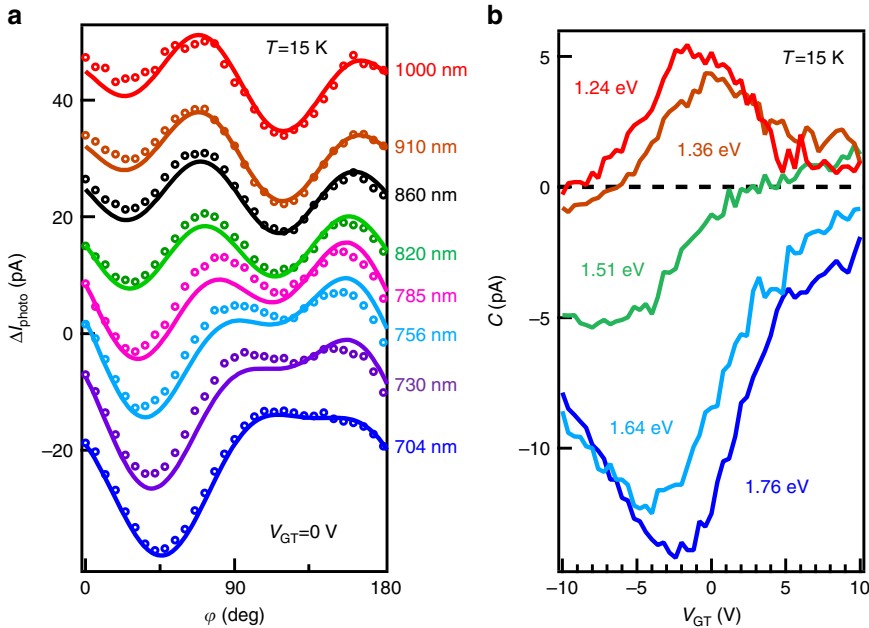

**Fig. 3** Photocurrent excited by photons of different wavelength, ranging from 704 nm (1.76 eV) to 1000 nm (1.24 eV). **a** The polarization-dependent photocurrent at each wavelength, with *solid lines* showing fits to Eq. (1). The curves are vertically offset for clarity. **b** The gate voltage dependence of the HDPC at five different photon energies ranging from 1.24 to 1.76 eV. At 1.24 eV, the HDPC is positive and reaches a peak at the voltage corresponding to the Dirac point in the $R_{xx}$ measurement. When we increase the photon energy, the HDPC starts to become negative and finally becomes a valley at 1.76 eV. The valley bottom is also around the Dirac point

drag effect[28]. Apart from $D$, $L_1$ and $L_2$, the HDPC identified by $C$ dominates the polarization-dependent photocurrent.

We now address the physical origin of the HDPC. Up to second order in the laser electric field, the HDPC can be attributed to two principal mechanisms, namely the CPGE and the circular photon drag effect (CPDE)[19, 20, 29]. Macroscopically, the CPGE is described by the expression $\mathbf{I}_j = \gamma_{js}(i\mathbf{e} \times \mathbf{e}^*)_s J$ and the CPDE is given by $\mathbf{I}_j = \tilde{T}_{jks}\mathbf{q}_k(i\mathbf{e} \times \mathbf{e}^*)_s J$, where $j, k, s = x, y, z$ and $\mathbf{e}$, $\mathbf{q}$ and $J$ denote the unit vector of the electric field, the momentum of the incident light and its intensity, respectively. The term $(i\mathbf{e} \times \mathbf{e}^*)$ identifies the helicity of the light, which varies from −1 to +1 and is zero for linearly polarized light. The non-zero components of the tensors $\gamma$ and $\tilde{T}$ are restricted by the crystal symmetry. For the bulk of $(Bi_{1-x}Sb_x)_2Te_3$, $\gamma$ is zero because of inversion symmetry. However, for the surface of $(Bi_{1-x}Sb_x)_2Te_3$ with a reduced symmetry of $C_{3v}$, a detailed symmetry analysis (Supplementary Note 1) suggests that the CPGE photocurrent along the $y$-axis is proportional to $\sin(2\varphi)\sin(\theta)$, while that from CPDE is proportional to $\sin(2\varphi)\sin(2\theta)$. Therefore, even though both effects have the same $\varphi$ dependence and contribute to the coefficient $C$ in Eq. (1), they can be differentiated through their distinct dependences on the incident angle $\theta$. Figure 1d shows a linear dependence of the coefficient $C$ with respect to $\sin\theta$, thus confirming the CPGE as the physical mechanism of the HDPC and ruling out any observable contributions from CPDE.

**Electrostatic gate tuning of the HDPC.** We next explore how the HDPC depends on the chemical potential and photon energy, with the aim of elucidating the microscopic origin of the HDPC and the role (if any) of the topological surface states. The chemical potential of $(Bi_{1-x}Sb_x)_2Te_3$ is tuned by a top gate voltage in a metal-oxide-TI heterostructure, as shown in Fig. 2a. The fabrication details of the heterostructure are described in Methods. We note that, in comparison with previous experiments wherein the chemical potential was tuned by controlling the chemical composition[30], electrostatic gating has the advantage of tuning the chemical potential continuously without changing the band structure. The photocurrent measurement is carried out in a 10 QL $(Bi_{0.5}Sb_{0.5})_2Te_3$ thin film—device B—at 15 K in a similar configuration as discussed above (Fig. 1a), with a fixed incidence angle of 33°. The sample has a chemical potential located in the bulk band gap, revealed by the insulating temperature dependence of the longitudinal resistance $R_{xx}$ (Supplementary Fig. 2). Figure 2b shows the photocurrent as a function of $\varphi$ for different gate voltages, along with fits to Eq. (1), shown by the *solid lines*. (The photocurrent at different gate voltages is offset on the $y$-axis for clarity.)

The gate voltage dependence of the HDPC, characterized by the coefficient $C$, is shown by the *blue line* in Fig. 2c. The plot also shows the longitudinal resistance, $R_{xx}$, as a function of the gate voltage $V_{GT}$ (*red line*). We note two interesting features in this plot. First, both the HDPC and $R_{xx}$ reach a maximum at the same gate voltage, when the chemical potential for electrons reaches the Dirac point. Second, the gate voltage dependence of the HDPC is asymmetric: it decreases faster with a positive gate voltage when the chemical potential is tuned toward the conduction band edge. These characteristics appear to be generic and are reproduced in device C (10 QL $(Bi_{0.54}Sb_{0.46})_2Te_3$ thin film) and device D (10 QL $(Bi_{0.57}Sb_{0.43})_2Te_3$ thin film), which were fabricated under similar conditions but with lower Sb concentrations (Supplementary Fig. 3).

Additional insights into the HDPC are gained by varying the wavelength of the optical excitation. Figure 3a shows the photocurrent as a function of polarization for several optical excitation wavelengths between 704 and 1000 nm, at a fixed excitation intensity. We extract the coefficient $C$ by fitting the data to Eq. (1). Figure 3b shows the gate voltage dependence of $C$ for different wavelengths, revealing a systematic pattern wherein the HDPC reverses sign as the wavelength is increased from 704 nm (photon energy of 1.76 eV) to 1000 nm (photon energy of 1.24 eV). Note that at the extreme ends of the wavelengths

studied (704 and 1000 nm), the magnitude of the HDPC still reaches a maximum close to zero gate voltage (i.e. when the chemical potential is at the Dirac point). At an intermediate wavelength, 820 nm (photon energy of 1.51 eV), the HDPC at zero gate voltage is close to zero and varies monotonically with the gate voltage without revealing any peak. The complex behavior of the HDPC when varying the gate voltage and light wavelength cannot be explained by a simple picture and requires a sophisticated theoretical model, which we develop below.

**Theoretical model of the HDPC.** Since a HDPC can only arise in the presence of broken inversion symmetry and the bulk crystal structure of tetradymite Bi-chalcogenides is inversion symmetric, previous studies of HDPC in 3D TIs have attributed its origin to topological surface states[31, 32]. However, such interpretations have yet to explain experimental observations rigorously. For instance, the photon energies used in past experiments probing HDPC in 3D TIs are in the range 1–2 eV, well above the bulk band gap where the surface states are located. Thus, the optical excitation cannot simply involve transitions between spin-momentum-locked surface states alone; the relevant optical transitions must involve both topological surface states and bulk states. In this case, it is not clear how circularly polarized photons can excite charge carriers with a preferred spin and momentum. We approach the problem of calculating the photocurrent by starting with a first-principles calculation of the band structure in $(Bi_{0.5}Sb_{0.5})_2Te_3$, as shown in Fig. 4c. Since surface states are located inside the bulk band gap around zero energy in Fig. 4c, we focus on bulk states in the energy range of $\pm 1 \sim \pm 2$ eV around the bulk gap (denoted by the *blue-shaded area*), which are described by $\Gamma_6^{\pm}$ or $\Gamma_{4,5}^{\pm}$ representations. Then, we consider the contribution to the photocurrent from different bulk states, separately. In this section, we focus on the $\Gamma_6^{\pm}$ bulk bands, which provide the dominant contribution. We will discuss the contribution from bands $\Gamma_{4,5}^{\pm}$ in the Discussion section.

Based on Fermi's golden rule, the general expression of photocurrent induced by optical transitions[32–34] between the surface states and the bulk states is derived as (Supplementary Note 2)

$$\mathbf{J} = -\frac{2\pi e}{\hbar} \sum_{\mathbf{k},\langle\eta,\xi\rangle} \left( \tau_{pb}\mathbf{v}_{\mathbf{b},\mathbf{k},\eta} - \tau_{ps}\mathbf{v}_{s,\mathbf{k},\xi} \right) \left| \langle \phi_{b,\mathbf{k},\eta} | H_{\text{int}} | \phi_{s,\mathbf{k},\xi} \rangle \right|^2$$
$$\left( f^0_{s,\mathbf{k},\xi} - f^0_{b,\mathbf{k},\eta} \right) \delta \left( E_{b,\mathbf{k},\eta} - E_{s,\mathbf{k},\xi} - \hbar\omega \right),$$

(2)

where $\tau_{pb}$, $\tau_{ps}$ represents the relaxation time of excited carriers in the bulk band and in the surface states, respectively; $H_{\text{int}}$ describes the interaction between electrons and light, $\phi_{b,\mathbf{k},\eta}$ and $E_{b,\mathbf{k},\eta}$ are the eigen-wavefunction and eigen-energy for bulk states with band index $b$ and spin index $\eta$, and $\phi_{s,\mathbf{k},\xi}$ and $E_{s,\mathbf{k},\xi}$ are for topological surface state, with the index $\xi$ labelling upper and lower Dirac cones. $\mathbf{v}_{b,\mathbf{k},\eta}$ and $\mathbf{v}_{s,\mathbf{k},\xi}$ are the velocity of bulk and surface bands, while $f^0_{b,\mathbf{k},\eta}$ and $f^0_{s,\mathbf{k},\xi}$ are the corresponding equilibrium Fermi distribution functions. Equation (2) can be evaluated numerically to obtain the photocurrent for photons with a certain helicity, but we first analyze the structure of this expression to identify the requirement for a non-zero photocurrent. The velocity operator $\mathbf{v}_{b(s),\mathbf{k},\eta(\xi)}$ in Eq. (2) is odd in $\mathbf{k}$, and thus only asymmetric terms of the transition matrix element $\left| \mathcal{M}_{\eta\xi} \right|^2 = \left| \langle \phi_{b,k,\eta} | H_{\text{int}} | \phi_{s,k,\xi} \rangle \right|^2$ with respect to $\mathbf{k}$ contribute to a non-zero photocurrent. The light-matter interaction $H_{\text{int}}$ in the matrix element is derived from the minimal coupling $\left( \mathbf{k} \to \Pi = \mathbf{k} - \frac{e}{\hbar}\mathbf{A} \right)$ as $-\frac{e}{\hbar}\frac{\partial H}{\partial k} \cdot \mathbf{A}$ and approximated to $-\frac{e}{m}\mathbf{A} \cdot \hat{\mathbf{p}}$ for a $\mathbf{k} \cdot \mathbf{p}$ Hamiltonian in the small momentum $k$ limit. For the optics setup shown in Fig. 1a, the vector potential of the light can be expressed as

$\mathbf{A}(t) = A_E e^{-i\omega t + i\mathbf{q}\cdot\mathbf{r}}(-\cos(\theta)(1 - i\cos(2\varphi)), \quad -i\sin(2\varphi), \quad -\sin(\theta)$ $(1 - i\cos(2\varphi))) + \text{c.c.}$, where $\varphi = \frac{\pi}{4}, \frac{3\pi}{4}$ corresponds to left, right circularly polarized light, respectively. The asymmetric part of the corresponding matrix element derived in Supplementary Note 3 is given by $\left( |\mathcal{M}_{\xi}|^2 \right)_a = \zeta_b \Delta\xi \sin(2\varphi)\sin(\theta)\sin(\theta_k)$, where the subscript $a$ labels the asymmetric part, $\theta_k$ identifies the angle of the momentum $\mathbf{k}$ with respective to the $k_x$ direction, $\zeta_b = +$ for valence bands and $\zeta_b = -$ for conduction bands; the parameter $\Delta$ depends on the bulk index $b$ and we sum over spin index $\eta$ of bulk bands because of the spin degeneracy. Thus, the generated photocurrent is proportional to $\sin(2\varphi)\sin(\theta)$, corresponding to the HDPC. For optical transitions from the valence band to the surface states ($v \to s$), we plot $\left( |\mathcal{M}_{v \to s,\xi}|^2 \right)_a$ as a function of $\theta_k$ in a polar graph in Fig. 4a for left circularly polarized light ($\varphi = \frac{\pi}{4}$) and assume $\Delta > 0$. The *red* (*blue*) color denotes the positive (negative) sign of $\left( |\mathcal{M}_{v \to s,\xi}|^2 \right)_a$. We notice that for the upper Dirac cone (UDC) ($\xi = +$), $\left( |\mathcal{M}_{v \to s,+}|^2 \right)_a > 0$ for positive $k_y$ and $\left( |\mathcal{M}_{v \to s,+}|^2 \right)_a < 0$ for negative $k_y$, leading to an asymmetric optical transition dominated by UDC electrons with positive $k_y$. In contrast, due to the sign reverse of $\left( |\mathcal{M}_{v \to s,-}|^2 \right)_a$ for the lower Dirac cone (LDC), optical transition is dominated by LDC electrons with negative $k_y$. Since the Fermi velocity also reverses sign between UDC and LDC for a fixed $k_y$, the optical transition is always dominated by the right-moving branch of surface electrons with a positive $y$-direction velocity, as shown schematically by the *red* coloring in the surface Dirac cones in Fig. 4a. Similarly, the optical transition from the surface states to conduction bands is dominated by the left-moving branch of surface electrons with a negative velocity, as shown in Fig. 4b.

We then perform a numerical simulation based on Eq. (2) for the photocurrent induced by optical transitions between topological surface states and two bulk bands (one conductance and one valence band with $\Gamma_6^+$ representation). $\Delta$ is assumed to be positive and the mass of the valence band holes is chosen to be heavier than that of the conduction band electrons. We used the same momentum relaxation time for the excited carriers in bulk states and in surface states for simplicity. (More details of the numerical calculation are described in Methods.) Our numerical calculations reveal the key feature observed in the experimental gate voltage dependence of the HDPC, namely, an asymmetric peak whose position is located at the Dirac point. We further separate the photocurrent $J$ into two parts: the current induced by the non-equilibrium distribution of carriers in the surface bands, denoted as $J_s$, and that induced by excited carriers in bulk bands, denoted as $J_b$ (*green* and *blue curves*, respectively, in Fig. 4d). The photocurrent contributed from conduction bands and valence bands are further separated in Supplementary Fig. 4. Interestingly, we find that $J_s$ depends monotonically on the chemical potential, while $J_b$ has a peak. This suggests that the peak mainly originates from the bulk photocurrent, while the asymmetry originates from the combined contribution of both topological surface states and bulk states. The latter arises because of the difference in mass between valence band holes and conduction band electrons.

## Discussion

For an intuitive understanding of the peak in the gate voltage dependence of the HDPC, we use a schematic picture of the optical transitions along the $k_y$ direction (Fig. 5), where (a), (b), (c) correspond to transitions with different positions of the chemical potential. The *circles* located in each band refer to the excited carriers that contribute to the net photocurrent. The mass difference between electrons and holes is neglected in the schematic picture for simplicity. Because preferential transitions occur on different branches of the surface states for $v \to s$ and $s \to c$, the

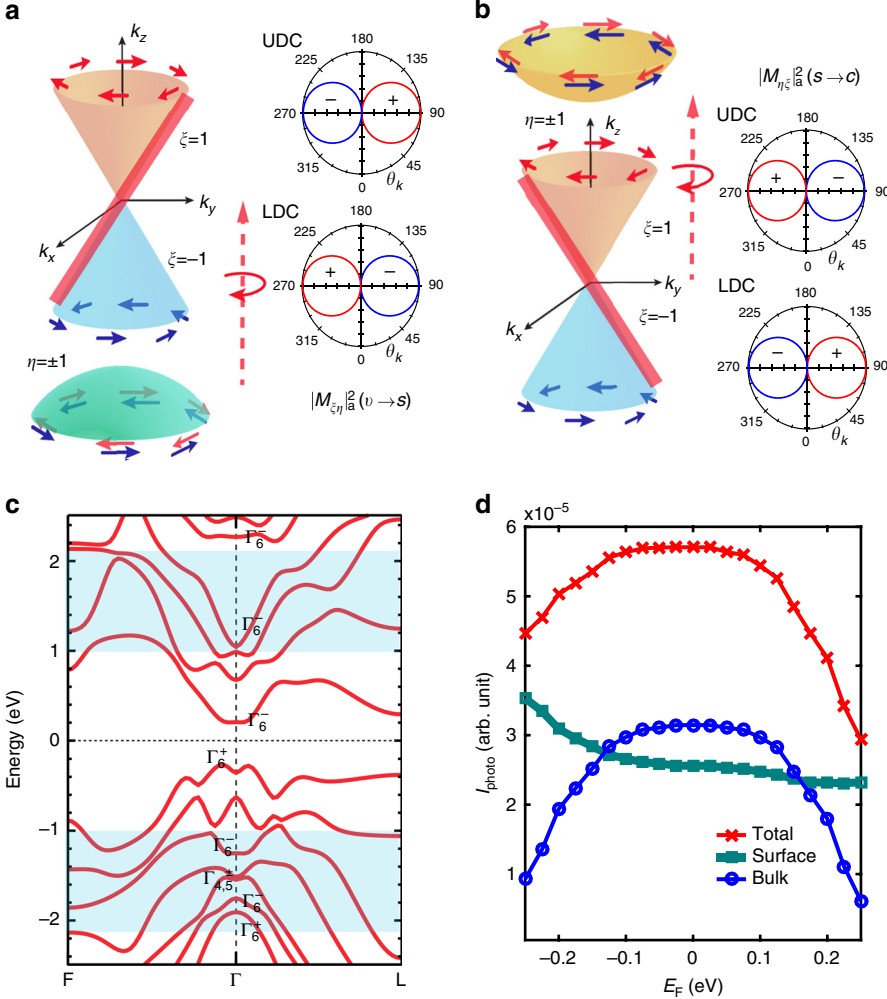

**Fig. 4** Theoretical analysis of optical transitions and numerical calculation of photocurrent. **a** Schematic picture of optical transitions from the bulk valence band to the Dirac surface states. $\xi$ and $\eta$ denote the spin texture of the surface states and the bulk states, respectively. The asymmetric matrix elements $\left(|\mathcal{M}_{\xi\eta}|\right)^2_a$ for the transitions are plotted in the polar graph of momentum angle ($\theta_k$) for upper and lower Dirac cones (UDC and LDC), where the sign is denoted by the color (*red* denotes positive and *blue* denotes negative). The asymmetric matrix element is positive on the positive $k_y$ side for the UDC, while positive on the negative $k_y$ side for the LDC. Therefore, the preferred optical transitions are from the valence band to the right-moving branch of the surface states, denoted by the *red coloring* on the Dirac cone. **b** Schematic picture of optical transitions from the Dirac surface states to the conduction band. The asymmetric matrix element is opposite compared to **a**. Therefore, the preferred optical transitions are from the left-moving branch of the surface states to the conduction band. **c** First-principle calculation of the bulk band structure of $(Bi_{0.5}Sb_{0.5})_2Te_3$. The irreducible representation to which each bulk band belongs is marked on top of the band. The *blue-shaded area* identifies the energy range, which is $\pm 1 - \pm 2$ eV away from the surface Dirac cone, relevant for optical transitions at the photon energy we use. **d** Calculated photocurrents based on the slab model as a function of the chemical potential. $E_F = 0$ eV denotes the chemical potential located at the Dirac point. The photocurrent contributions from the excited carriers in the surface states (*green curve*), the bulk bands (*blue curve*) and the total photocurrent (*red curve*) are displayed separately

excited electrons (*filled blue circles*) and holes (*empty red circles*) in the surface states induce a photocurrent along the same direction. The comparison between (a), (b) and (c) shows that the photocurrent contribution from the surface states changes little with the chemical potential. However, the bulk states contribute to the photocurrent in a very different manner. When the chemical potential is located at the Dirac point (Fig. 5b), the photocurrent contribution from the bulk bands (identified by *empty blue circles* and *filled red circles*) is maximized along the negative *y*-direction, denoted by the *red and blue arrows*. As the chemical potential moves away from the Dirac point, either down toward the valence band or up toward the conduction band as shown in Fig. 5a, c, respectively, the photocurrent contributed from different bulk bands partially cancels out and decreases (see inset to Fig. 5). This physical picture accounts for the HDPC reaching a maximum value at the Dirac point.

The asymmetry of the HDPC around the Dirac point (shown in Fig. 2c) can be understood from the following three reasons. First, our numerical simulation has revealed the asymmetry of the HDPC (shown in Fig. 4d), which is induced by the heavier mass of valence band holes, compared to that of conduction band electrons. Second, the transition matrix element between the surface states and the conduction bands is different from that between the surface states and the valence bands. Third, first-principles calculations (Fig. 4c) show the absence of conduction band states located at 1.96 eV above the band gap for $(Bi_{0.5}Sb_{0.5})_2Te_3$. Thus, with a photon excitation of 1.96 eV, the HDPC is dominated by optical transitions from the valence band to the surface states ($v \rightarrow s$). As the $v \rightarrow s$ optical transitions become forbidden by populating the surface states with a positive gate voltage, the photocurrent decreases faster. Last, as we use different relaxation times for excited carriers in surface states,

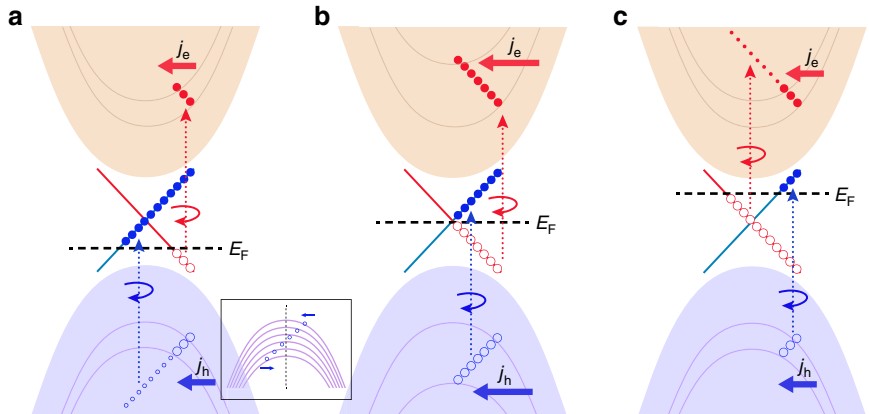

**Fig. 5** Schematic picture to explain the chemical potential dependence of the HDPC. **a–c** denote the chemical potential below, near and above the Dirac point, respectively. The bulk bands are assumed to be continuous in the energy range of the photons and the surface states are perfect Dirac bands. The *red dotted* arrows denote the preferred transitions from the surface states to conduction bands ($s \rightarrow c$), while the *blue dotted arrows* denote the preferred transitions from the valence bands to surface states ($v \rightarrow s$). The two transitions happen simultaneously and rely on the position of the chemical potential. **a** The chemical potential is below the Dirac point. The excited carriers are denoted by *filled circles* (electrons) and *empty circles* (holes). In the valence band, the excited holes are separated into two parts, one denoted by *small circles* and the other denoted by *large circles*. As shown in the sub-diagram, the small circles are distributed evenly around the $\Gamma$ point, leading to zero total generated photocurrent. Therefore, only the *large circles* contribute to the net photocurrent. The net photocurrent induced by the bulk carriers is along the negative *y* direction and denoted by the *arrows* pointing to the left. The length of the *arrows* denotes the magnitude of the photocurrent. **b** The chemical potential crosses the Dirac point. The length of the *arrows* reach a maximum indicating the largest photocurrent. **c** The chemical potential is above the Dirac point. The excited carriers in the conduction band are separated into two parts and the *small circles* do not contribute a net photocurrent. Therefore, the length of the *arrows* denoting the net photocurrent decreases

valence bands and conduction bands in the numerical calculation (Supplementary Figs. 5, 6 and Supplementary Note 4), the asymmetry of the calculated HDPC around the Dirac point is tuned by different relaxation times.

We next comment on the wavelength dependence of the HDPC, based on the band structure of $(\text{Bi}_{0.5}\text{Sb}_{0.5})_2\text{Te}_3$ in Fig. 4c. Figure 3 shows that the HDPC reverses sign as the photon energy is varied from 704 nm (photon energy of 1.76 eV) to 1000 nm (photon energy of 1.24 eV). To understand this sign reversal, we note that in Fig. 4c the bulk valence bands change from $\Gamma_6^-$ to $\Gamma_{4/5}^{\pm}$ to $\Gamma_6^+$ at an energy about 1.5 eV lower than the bulk band gap where the surface states are located. Thus, the HDPC is expected to sensitively depend on photon energies through the optical transitions from different bulk valence bands to surface states. Our analysis in Supplementary Note 3 shows that the optical transition between the $\Gamma_{4/5}^{\pm}$ bands and the surface states will not contribute to the photocurrent. Our theoretical formulae (Supplementary Note 3) suggest that the parameter $\Delta$ in the transition matrix element should have opposite signs for the $\Gamma_6^-$ and $\Gamma_6^+$ bands. The sign reversal of $\Delta$ induces the sign reversal of the HDPC at 1.5 eV.

Finally, we examine the possibility of contributions to the HDPC from other types of spin-split states, specifically Rashba states induced by band bending[26] and the second excited surface states that were experimentally discovered by ARPES[35]. If we only consider the Rashba effect in the conduction bands (because of the lighter mass), we find that the Rashba effect contributes to an extra photocurrent, which is proportional to $-\alpha_R(A_0^2 k_c^2 - \text{sgn}(E_F)E_F^2)\hat{e}_y$, where $\alpha_R$ is the strength of the Rashba effect, $A_0$ is the coefficient of the surface states' Dirac Hamiltonian and $k_c$ is the cut-off momentum (Supplementary Note 3). This expression implies that the contribution from the Rashba effect, if it exists, only contributes to the asymmetry of the HDPC peak around the Dirac point. The second excited surface states[18, 27] are located around 1.5 eV above the first Dirac point, and thus the optical transition between the two Dirac cones in our photon energy range should exist. This optical transition between the two Dirac cones could contribute to the HDPC according to the analytical study of the photocurrent in Supplementary Note 5. However, we also note from the analytical study that the HDPC from the transitions between the two Dirac cones should increase monotonically as the chemical potential moves up in the gap, which contradicts with our experimental observation of the gate-dependent photocurrent. Further, we rule out another possible origin of the HDPC, namely the surface shift current[23], by studying the in-plane azimuthal angle dependence of the HDPC (Supplementary Fig. 7). Finally, the photocurrent imaging with a 5 μm-wide He-Ne laser shows no spatial gradient of HDPC induced by laser heating (Supplementary Fig. 8).

In summary, our systematic measurements provide insights into the HDPC in a 3D TI (Sb-doped $\text{Bi}_2\text{Te}_3$). By studying the incidence angle dependence, we unambiguously demonstrate that the CPGE is the physical mechanism underlying the HDPC, and we rule out the photon drag effect. Further, the gate voltage dependence experiment reveals a maximum of the HDPC as the chemical potential approaches the Dirac point. A theoretical model is constructed to qualitatively explain the chemical potential dependence of the HDPC and to identify the microscopic origin of the asymmetry of the HDPC when the Fermi energy is tuned above or below the surface Dirac point. The sign change of the HDPC when varying excitation photon energy around 1.5 eV is attributed to different contributions of photocurrents from the bulk $\Gamma_6^+$ band and $\Gamma_6^-$ band. The demonstration of a robust directional photocurrent that can be systematically controlled by the polarization and wavelength of light and tuned by an electrostatic gate voltage creates interesting new opportunities for using 3D TIs in opto-spintronics[36–38].

## Methods

**Sample fabrication.** We grew $(\text{Bi}_{1-x}\text{Sb}_x)_2\text{Te}_3$ thin films on sapphire substrates by molecular beam epitaxy. A 1–2 QL $\text{Bi}_2\text{Se}_3$ thin film was first deposited as a seed layer, followed by the growth of a $(\text{Bi}_{1-x}\text{Sb}_x)_2\text{Te}_3$ epilayer of desired thickness. The Bi:Sb ratio allowed us to adjust the chemical potential of the as-grown sample into the bulk band gap[39]. After growth, the samples were capped in situ with 4 nm Al, which oxidizes upon exposure to ambient atmosphere and protects the sample surface. We made device A from a 20 QL thin film a and devices B, C and D from 10 QL thin films b, c and d, respectively. We used x-ray photoelectron

**Table 1 Parameters used for the model study of $(Bi_{1-x}Sb_x)_2Te_3$**

| $M_0$ (eV) | $M_1$ (eV·Å²) | $A_0$ (eV·Å) | $B_0$ (eV·Å) | $\alpha_R$ (eV·Å) |
|---|---|---|---|---|
| −0.28 | 6.86 | 3.33 | 2.26 | 0 |
| =x(112)$A_3$ (eV·Å) | =x(112)$B_3$ (eV·Å) | $E_c$ ($E_v$) (eV) | =x(112)$M_c$ (eV·Å²) | =x(112)$M_v$ (eV·Å²) |
| 1.66 | 1.13 | 1.5(−1.5) | 50 | 30 |

spectroscopy to identify the Bi, Sb ratio of thin film b, and obtained the Bi, Sb ratio of the other films from the Bi, Sb flux of the MBE growth based on that of thin film b. We used two techniques to pattern electrical channels for the photocurrent measurements: mechanical patterning using a thin metal tip and an automated stage to scratch away the thin film. This was applied to device A, on which we patterned a 0.5 mm × 1 mm channel. Devices B, C and D were patterned into 0.1 mm × 0.5 mm channels using optical lithography. For these devices, we also fabricated a top gate by first removing the protective Al capping layer and then depositing 30 nm $Al_2O_3$ by atomic layer deposition, followed by a sputter-deposited 200 nm layer of indium tin oxide (ITO). The removal of the Al capping layer is to improve the quality of the deposited $Al_2O_3$, which serves as the dielectric layer. The transparent ITO serves as the top gate electrode. For all the samples, we used In dots to make ohmic contacts to the films.

**Photocurrent measurement with polarization control.** The photocurrent was excited by 7.5 mW of 633 nm excitation from a continuous He-Ne laser modulated by an optical chopper at 569 Hz. To eliminate any contribution from the photo-thermoelectric effect from the surface states[40], the laser was obliquely incident on the center of the channel with a 100 μm beam diameter. We changed the polarization of the beam by rotating a quarter wave plate. The photocurrent was measured along the y-axis, perpendicular to the incidence plane of the laser beam. We grounded one end of the conduction channel and used a virtual ground current-to-voltage preamplifier and a lock-in amplifier to detect the photocurrent. The wavelength dependence of the photocurrent was studied using a Ti-sapphire laser, with the intensity kept constant at 7 mW.

**Numerical calculation of the photocurrent.** We used an eight-band effective Hamiltonian for the numerical calculation, which includes a four-band model $H_1$ for the description of topological surface states and two doubly degenerate bulk bands. Explicitly, $H_{8×8} = H_1 \otimes H_2$ with $H_1 = (M_0 + M_1 k_z^2)\tau_z \otimes \sigma_0 + B_0 k_z \tau_y \otimes \sigma_0 + A_0 \tau_x \otimes (\sigma_x k_y - \sigma_y k_x)$ on the basis $(|P_1^+,\uparrow\rangle, |P_1^+,\downarrow\rangle, |P_2^-,\uparrow\rangle, |P_2^-,\downarrow\rangle)^T$ from $\Gamma_6^\pm$ irreducible representation and $H_2 = ((E_c + M_c\mathbf{k}^2)\sigma_0 + \alpha_R(k_y\sigma_x - k_x\sigma_y)) \otimes \frac{1}{2}(\tau_0 + \tau_z) + (E_v - M_v\mathbf{k}^2\sigma_0) \otimes \frac{1}{2}(\tau_0 - \tau_z)$ on the basis $(|P_3^+,\uparrow\rangle_c, |P_3^+,\downarrow\rangle_c, |P_3^+,\uparrow\rangle_v, |P_3^+,\downarrow\rangle_v)^T$ from $\Gamma_6^+$ irreducible representation, where subscript $c(v)$ denotes conduction band(valence band) and $\sigma_{0,x,y,z}(\tau_{0,x,y,z})$ are unit/Pauli matrices, representing spin (orbital) space. The zeroth order of the light-matter interaction Hamiltonian in the small $k$ limit on the basis $(|P_2^-,\uparrow\rangle, |P_2^-,\downarrow\rangle, |P_3^+,\uparrow\rangle, |P_3^+,\downarrow\rangle)^T$ is expressed as $H_{int} = \frac{e}{\hbar}(B_3 A_z \tau_y \otimes \sigma_0 + A_3 A_x \tau_x \otimes \sigma_y - A_3 A_y \tau_x \otimes \sigma_x)$. We construct a slab model by transforming the continuous model to a lattice model through $k \to \frac{\sin(ka)}{a}$ and $k^2 \to \frac{2(1-\cos(ka))}{a^2}$, where $a = 5$ Å can be regarded as the lattice constant. Thirty quintuple layers are considered in the slab model calculation. The velocity of carriers is taken as $\mathbf{v} = \frac{1}{\hbar}\frac{\partial H}{\partial \mathbf{k}}$. The delta function $\delta(E_F - E_i - \hbar\omega)$ in the photocurrent formula is replaced by $\frac{1}{\sqrt{2\pi}\Gamma_0} e^{-\frac{(E_F - E_i - \hbar\omega)^2}{2\Gamma_0^2}}$, where $\Gamma_0 = 0.3$ eV denotes the energy band width due to thermal fluctuation and disorder. The relaxation time of the excited carriers is taken as 1 ps[41] in our numerical calculation and $\theta = \frac{\pi}{4}$, $\varphi = \frac{\pi}{4}$ are chosen for the vector potential. The amplitude of vector potential combined with $\frac{e}{\hbar}$ is taken as $\frac{e}{\hbar}A_E = 0.05$ Å⁻¹. The photon energy we use is 1.96 eV. The other parameters we use are listed in Table 1.=x(103)

**Data availability.** The dataset that supports the findings of this study is available from the corresponding author upon reasonable request. Details of the analytical calculation are provided in the supplementary material.

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

## Acknowledgements

This work was supported by grants from ONR (Nos. N00014-15-1-2369, N00014-15-1-2370 and N00014-15-1-2675), NSF (DMR-1306510, DMR-1306300) and ARO MURI (W911NF-12-1-0461). This study is based in part on research conducted at The Pennsylvaniastate University 2D Crystal Consortium–Materials Innovation Platform(2DCC-MIP), which is supported by NSF Cooperative AgreementDMR-1539916. We also acknowledge partial support from DARPA and C-SPIN, one of six centers of STARnet, a Semiconductor Research Corporation program, sponsored by MARCO and DARPA.

## Author contributions

Y.P. designed and performed the experiments, collected/analyzed the data and wrote the manuscript. Q.-Z.W. performed the theoretical analysis and numerical calculations of the photocurrent and wrote part of the paper and the supplementary. A.L.Y. and D.D.A. contributed to measurements of the wavelength dependence of the photocurrent experiment, to the photocurrent imaging in the supplementary, and to editing of the manuscript. T.P. performed the experiments and collected the data. T.C.F. and A.R. grew the samples for this study. H.Z. performed the first principles calculation of the bulk band structure of $(Bi_{1-x}Sb_x)_2Te_3$. C.-X.L. contributed to the theoretical study of the photocurrent and edited the manuscript. N.S. conceived the experiment, contributed to every stage of the experiments, and edited the manuscript. All authors discussed the results and commented on the manuscript.

## Additional information

**Competing interests:** The authors declare no competing financial interests.

