## [Peer Review File · Nature Communications]

Reviewers' Comments:

Reviewer #1 (Remarks to the Author):

The manuscript by Yu Pan et al. reports on the observation of the circular photogalvanic effect in Sb₂Te₃ films. The important conclusion is that the dc current generated by light originates from the surface electrons only, as the inversion symmetry of bulk Sb₂Te₃ would filter out contributions from bulk electrons. Although such phenomenon is expected based on standard symmetry filtration rule in PGE, it is not straightforward to experimentally confirm the dominant contribution from topological surface states. In particular, studying the angle of incidence dependence of the photocurrent allowed excluding the contribution of the circular photon drag effect, which can be generated in both surface states and the bulk. The experimental data are convincing, since various control experiments were performed to narrow-down the origin of the effect based on expected presence or lack of symmetries in surface and bulk. Hence the paper opens an avenue to study symmetry-induced or topologically protected surface states features being of substantial present interest. The manuscript and supplementary material describe the experiments and analysis well. The paper is of high scientific quality, both experimentally as well as theoretically. By this combination of content and technique the paper thus represents a noteworthy advance in the study of new quantum states of matter, so publication in Nature Communications is strongly recommended.

There are, however, a few shortcomings concerning detailed aspects of the work as presented, which the authors should address.

1. Discussing the circular photon drag effect the authors cited the book of Sturman and Fridkin “Photogalvanic effects” (Ref. 27). This book is aimed to photogalvanic effects only and does not address the photon drag effects. Therefore, I suggest replacing the reference 27 by the first theoretical paper predicting this effect and two experimental articles reporting on its observation in 2D systems with and without spin-orbit coupling.

- V.I. Belinicher, Fiz. Tverd. Tela 23 3461 (1981) [Sov. Phys. Solid State 23, 2012 (1981)]
- V. A. Shalygin et al. JETP Lett. 84, 570 (2006).
- C. Jiang et al., Phys. Rev. B 84, 125429 (2011).

2. The theoretical model and calculations described in the manuscript suggest that the momentum relaxation time τ_p is independent of energy. Typically, τ_p is energy-dependent

and is different for different bands. This can give rise to additional contributions to photocurrent, e.g., at the optical transitions between the first and the second surface states. Similar effect for quantum wells has been addressed by S.D. Ganichev et al., Resonant inversion of the circular photogalvanic effect in n-doped quantum wells, Phys. Rev. B 68, 035319 (2003). A corresponding comment or justification of the approximation used would be useful.

3. While being well defined I found that the used notations for the tensors and angles unnecessarily complicate the reading because they deviate from the terms typically used in literature (see e.g. E.L. Ivchenko, OPTICAL SPECTROSCOPY OF SEMICONDUCTOR NANOSTRUCTURES, Alpha Science International Ltd 2005 and a number of other books on photogalvanics). In particular, using θ for the quarter plate rotation angle and φ for the angle of incidence slow down the reading, because in most of the papers and books on this topic it is just vice versa. May the authors can consider using θ for the angle of incidence, φ for the quarter plane rotation angle, γ for the CPGE, \tilde{T} or T for the CPDE and χ for the linear PGE.

Reviewer #2 (Remarks to the Author):

The manuscript by N. Samarth et al report a comprehensive study of the helicity-dependent photocurrent in a typical topological insulator material $(\text{Bi}_{1-x}\text{Sbx})_2\text{Te}_3$ thin films as a function of the incidence angle of the optical excitation, the excitation wavelength and the gate-tuned chemical potential. The experimental observation confirms that the photo-galvanic effect nature of the observed helicity-dependent photocurrent response. By combining with the first-principles numerical calculation of the photocurrent, they further conclude that the helicity-dependent photocurrent originates from the asymmetric optical transitions between the topological surface states and bulk. The experimental results are quite interesting, the interpretation based on their numerical calculation is consistent with all the observed features and is thus convincing, which brings new insight for the observed helicity-dependent photocurrent in topological insulator materials. I therefore recommend publication of this work on Nature Communication. However, I would like the authors address the following two issues:

1. Since the photoexcited electron density distribution Δp is related to the optical transition process, when authors do the first-principles numerical calculation of the photocurrent, how do they treat all the carrier relaxation processes that could be involved in between different multiple bands, when they tune the excitation photon energy? I noticed that they took the relaxation time τ_p as 0.1ps, does this mean that τ_p is always treated as a constant?
2. As illustrated in figure 4a, 4b, the authors calculated the photoexcited electron density distribution Δp related to the optical transition processes from the bulk valence band to the

surface states, and from the surface states to the bulk conduction band. From their numerical calculation, it is curious to know that if the authors find different features for the helicity-dependent photocurrent resulting from the above two optical transition processes,? It seems expected that the optical transition process from bulk valence band to the surface states may exhibit different feature due to the complexity and asymmetry of the valence band.

Reviewer #3 (Remarks to the Author):

The manuscript gives a very thorough study of photocurrents generated in topological insulator films. In this manuscript, the photocurrents are generated by transitions from/to the bulk bands to the surface states. The authors give evidence that the mechanism that dominates the generation of the photocurrent is the circular photo-galvanic effect. They also find a non-monotonic behavior of the magnitude of the photocurrent as a function of the top gate voltage (which is supposed to tune the Fermi level in the surface states), as well study the dependence of the photocurrent on the energy of the incident photons.

Regarding the suitability of the manuscript for Nature Communications. On one hand, the authors did a very comprehensive study of the photocurrent generation, and complemented the experimental work with a suitable theoretical analysis, which seems to explain most of the experimental findings. On the other hand, the results presented in this manuscript are not really surprising or demonstrate any fundamentally new effect. Moreover, many of the results, including item 2 and 3 of the main claims on page 3-4 of the manuscript, are probably not universal and very specific to the specific material studied in the current manuscript. Taking these points into account, it is my opinion that the thorough study performed in this work does merit publication in Nature Communications. However, the authors should address several technical points:

- 1) The evidence given in the manuscript that the Fermi level is indeed situated inside the bulk gap (and thus in the surface states) is not convincing. The peak in the resistivity presented in the manuscript can be a result of the bulk effect. Additional evidence that there is no bulk resistivity is required.
- 2) To unambiguously identify the effect as the circular photogalvanic effect, it is important to rule out the possibility that the effect arises due to an asymmetric heating of the sample due to incident light, which arises to the specific geometry of the sample and light spot. The authors show a linear fit to $\sin(\phi)$, however changing the angle ϕ might also change the geometry of the experiment, i.e. the illumination spot vs. the sample shape. The authors need to rule out this effect.
- 3) The data for the photocurrent shows a dominant component for which is independent of the polarization. I presume that this component is also independent of the angle ϕ , although this is

not clearly discussed. What is the origin of this component? Also, what is the origin of the nonzero components for linear polarization?

4) On page 9 and Figure 4, the authors discuss and present a calculation of the matrix element, squared, which seems to take both positive and negative signs. This is of course not possible, and the authors should explain what they mean here.

5) The authors should emphasize the new insights resulting from their experiment vs. those that can be learned from the experiment done by McIver et. al. (Ref 15).

6) The authors should discuss which of the detailed features of the photocurrent response (dependence on photon energy and on the Fermi level) are universal, and which are specific to the material used in this experiment.

Response to reviewers

Reply to the first reviewer:

1) Discussing the circular photon drag effect the authors cited the book of Sturman and Fridkin "Photogalvanic effects" (Ref. 27). This book is aimed to photogalvanic effects only and does not address the photon drag effects. Therefore, I suggest replacing the reference 27 by the first theoretical paper predicting this effect and two experimental articles reporting on its observation in 2D systems with and without spin-orbit coupling.

Reply: We have followed the suggestion and replaced reference 27 by the three articles that were pointed out by the referee.

2) The theoretical model and calculations described in the manuscript suggest that the momentum relaxation time τ_p is independent of energy. Typically, τ_p is energy-dependent and is different for different bands. This can give rise to additional contributions to photocurrent, e.g., at the optical transitions between the first and the second surface states. Similar effect for quantum wells has been addressed by S.D. Ganichev et al., Resonant inversion of the circular photogalvanic effect in n-doped quantum wells, Phys. Rev. B 68, 035319 (2003). A corresponding comment or justification of the approximation used would be useful.

Reply: As the referee indicates, the constant value of relaxation time τ_p is indeed a simplification in our theoretical model. In realistic systems, τ_p should be energy dependent and different for different bands. To understand the influence of energy dependent relaxation time, we performed numerical calculations of the photocurrent with different relaxation times for different bands (surface, conduction and valence bands), as shown in the updated Supplementary Note 4. Supplementary Figure 5 and 6 show the photocurrents as a function of Fermi energy for different relaxation times of surface states, conduction and valence bands. We find that different relaxation times only change the results quantitatively while all the essential qualitative features, such as the peak at the Dirac point and the asymmetry between the electron and hole doping regimes, remain the same. More quantitatively, by comparing the line shape of the calculated photocurrents with that in experiments, we conclude that a shorter surface relaxation time ($\tau_s = 0.5 \tau_b$ in Supplementary Figure 7) and a longer conduction band relaxation time ($\tau_c = 2 \tau_v$ in Supplementary Figure 8) give rise to a better match with the experimental observations. We note that the relaxation process for photo-excited carriers in topological insulators have been studied in several time-resolved measurements [1-6] and the relaxation time reported for excited bulk carriers is slightly longer than that of surface carriers[3][4], thus providing additional support for our conclusion.

The referee also mentions the possible contribution of photocurrents from the optical transitions between the first and the second surface states. Our analytical results in Supplementary Note 4 confirm this contribution of photocurrents, which is proportional to the difference in relaxation times between two surface states. However, we note that the induced

photocurrent will monotonically increase with chemical potential, contradicting our experimental observations. Therefore, the photocurrent contributed by the transitions between the two Dirac surface states is not the major origin of the HDPC we observe.

3. While being well defined I found that the used notations for the tensors and angles unnecessarily complicate the reading because they deviate from the terms typically used in literature (see e.g E.L. Ivchenko, OPTICAL SPECTROSCOPY OF SEMICONDUCTOR NANOSTRUCTURES, Alpha Science International Ltd 2005 and a number of other books on photogalvanics). In particular, using θ for the quarter plate rotation angle and φ for the angle of incidence slow down the reading, because in most of the papers and books on this topic it is just vice versa. May the authors can consider using θ for the angle of incidence, φ for the quarter plane rotation angle, γ for the CPGE, \tilde{T} or T for the CPDE and χ for the linear PGE.

Reply: We changed the notation for the angles, including the quarter wave plate rotation angle and the angle of incidence and the notation of the tensors for CPGE and CPDE as suggested.

Reply to the second reviewer:

1) Since the photoexcited electron density distribution $\Delta\rho$ is related to the optical transition process, when authors do the first-principles numerical calculation of the photocurrent, how do they treat all the carrier relaxation processes that could be involved in between different multiple bands, when they tune the excitation photon energy? I noticed that they took the relaxation time τ_p as 0.1ps, does this mean that τ_p is always treated as a constant?

Reply: This question was also raised by the first referee (Q2). Indeed, the constant relaxation time in our theoretical model is an approximation. We justify this approximation and study the effect of different relaxation times for different bands in the updated Supplementary Note 4. A detailed response is found in our reply to question 2 of the first referee.

2) As illustrated in figure 4a, 4b, the authors calculated the photoexcited electron density distribution $\Delta\rho$ related to the optical transition processes from the bulk valence band to the surface states, and from the surface states to the bulk conduction band. From their numerical calculation, it is curious to know that if the authors find different features for the helicity-dependent photocurrent resulting from the above two optical transition processes? It seems expected that the optical transition process from bulk valence band to the surface states may exhibit different feature due to the complexity and asymmetry of the valence band.

Reply: To address this question, we have added new calculations to the supplementary materials. The photocurrent contributions from transitions between the surface states and the bulk conduction band transitions and between bulk valence band states to the surface states

are separately shown in Supplementary Figure 4 (the blue line for valence bands and black line for conduction bands). We note that the overall features for these two contributions are quite similar and only the magnitude of the contribution from valence bands is larger than that from conduction bands due to the large density of states for valence bands. We emphasize that the situation for (Bi,Sb)₂Te₃ is different from that for conventional III-V or II-VI semiconductors. In the conventional III-V or II-VI semiconductors, the conduction band has s-orbital nature while the valence band has p-orbital nature, leading to a more complicated band structures for valence band compared to conduction band. For (Bi,Sb)₂Te₃, both conduction and valence bands are from the p orbitals of Bi, Sb and Te atoms[7], and thus, they are qualitative similar for the purposes of these calculations.

Reply to the third reviewer:

1) The evidence given in the manuscript that the Fermi level is indeed situated inside the bulk gap (and thus in the surface states) is not convincing. The peak in the resistivity presented in the manuscript can be a result of the bulk effect. Additional evidence that there is no bulk resistivity is required.

Reply: In addition to the gate voltage dependent longitudinal resistance R_{xx} , we have added the temperature dependent R_{xx} at zero gate voltage while cooling down the sample from room temperature (RT) to 15 K in the supplementary material (Supplementary Fig 2). We show that R_{xx} increases monotonically as the temperature cools down, which is a signature of an insulator. This temperature dependence is consistent with the insulating samples studied in Ref [8]. Thus, the temperature dependence of R_{xx} provides another evidence that the chemical potential at zero gate voltage is indeed in the bulk band gap.

2) To unambiguously identify the effect as the circular photogalvanic effect, it is important to rule out the possibility that the effect arises due to an asymmetric heating of the sample due to incident light, which arises to the specific geometry of the sample and light spot. The authors show a linear fit to $\sin(\phi)$, however changing the angle ϕ might also change the geometry of the experiment, i.e. the illumination spot vs. the sample shape. The authors need to rule out this effect.

Reply: We ran a scanning photocurrent measurement on the same piece of sample, shown in the Supplementary Figure 8. The photocurrent imaging shows that the circular photo-galvanic effect (CPGE) does not vary too much when we scan the beam across the sample covered under the gate. We also note that the beam size used in the scanning measurement is more than ten times smaller than the beam size for the incidence angle dependence measurement. For a larger beam size, the photocurrent variation with position change would be even smaller. Further, there is no signature of heating induced change of CPGE in the imaging result. (If there is CPGE induced by heating, we would expect to see a larger CPGE when the beam moves away from the center of the sample, which we did not see in our scanning photocurrent measurement.) Therefore, in the incidence angle dependence study, the slight change of the beam spot position when tuning the incidence angle should not affect the CPGE. Also, additional beam size change caused by the incidence angle variation should not deliver a sine

function dependence. Therefore, we are confident that the linear dependence on the sine function of the incidence angle is an effect from the physical origin of the CPGE.

3) The data for the photocurrent shows a dominant component for which is independent of the polarization. I presume that this component is also independent of the angle ϕ , although this is not clearly discussed. What is the origin of this component? Also, what is the origin of the nonzero components for linear polarization?

Reply: We do observe a polarization independent photocurrent and a linear polarization dependent photocurrent shown in Fig. 1(b). However, we would like to stress that the CPGE is still dominant in the total photocurrent, as shown in Fig. 1(c). We also find that the polarization independent photocurrent mainly originates in a dark current that we observe even without a laser, usually in the order of 2-10 pA. The dark current is not a real photocurrent generated by laser, and therefore, is not induced by an optical effect. Also, this polarization independent photocurrent varies little with the angle ϕ . The linear polarization dependent photocurrent, instead, is a real optical effect. People have already done experiments focusing on the linear polarization dependent photocurrent in topological insulators. They find that the origin of this photocurrent is possibly the photon drag effect [9][10]. We have added a few sentences in the main text to point out the origin of the other two photocurrent components.

4) On page 9 and Figure 4, the authors discuss and present a calculation of the matrix element, squared, which seems to take both positive and negative signs. This is of course not possible, and the authors should explain what they mean here.

Reply: The referee is correct that the matrix element, squared, should always be a positive number. Figure 4 does not show the full matrix element; instead, we split the matrix element into the symmetric and asymmetric parts with respect to the momentum. Figure 4 only shows the asymmetric part, which gives rise to the non-zero photocurrent and determines the direction of the photocurrent. We have changed the notation of the matrix element in the paper to make it clear.

5) The authors should emphasize the new insights resulting from their experiment vs. those that can be learned from the experiment done by McIver et. al. (Ref 15).

Reply: As discussed in detail in the introduction, our systematic study studies the helicity dependent photocurrent in topological insulators by tuning the incidence angle, the chemical potential and the wavelength of laser. This allows to address the microscopic origin of the helicity dependent photocurrent (HDPC). In addition, our demonstration of the maximal HDPC at the Dirac point suggests the important role of surface states for this phenomenon and opens a path to potential applications of topological surface states in opto-spintronics. While Ref. 15 described the elegant experimental discovery of the HDPC in topological insulators, the experiments in that manuscript did not propose a detailed explanation of the origin of the phenomenon.

6) The authors should discuss which of the detailed features of the photocurrent response (dependence on photon energy and on the Fermi level) are universal, and which are specific to the material used in this experiment.

Reply: The emergence of the peak in the HDPC as the chemical potential crosses the Dirac point is universal for materials that hold Dirac surface states and inversion symmetric bulk once a single bulk band dominates the contribution. However, the HDPC reversal at a photon energy around 1.5 eV is due to the contributions from multiple bulk states, and is thus not universal.

In materials with Dirac surface states and inversion symmetric bulk, the HDPC also originates from the asymmetric optical transitions between the Dirac surface states and the bulk states. Thus, similar matrix elements can be used to describe the optical transitions. The schematic picture of the optical transitions shown in Figure 5 also applies. Therefore, we would also expect a peak of HDPC at the Dirac point. However, the direction of the photocurrent ($\pm\hat{y}$) depends on several material-dependent parameters and the spin texture of the Dirac-cone surface states. The direction of the HDPC for topological insulators with right-handed spin texture would be opposite to what we have for $(\text{Bi, Sb})_2\text{Te}_3$.

The sign reversal of HDPC when we tune the photon energy is deeply connected to the bulk band structure and parity. As presented in the main text and Fig 4(c), the bulk band at the Γ point around -1.76 eV belongs to the Γ_6^- irreducible representation while the bulk band at the Γ point around -1.92 eV belongs to the Γ_6^+ irreducible representation. This leads to the sign reversal of the photocurrent at a critical photon energy around 1.5 eV. For a different material, the sign reversal might not appear in the same energy range due to the different bulk band property. However, the idea that the helicity dependent photocurrent may reverse the direction if the bulk band parity reverses holds true for other topological insulator materials. From this perspective, our photon energy dependence study of the HDPC is universal.

1. J.A. Sobota, S. Yang, J.G. Analytis, Y.L. Chen, I.R. Fisher, P.S. Kirchmann and Z.-X. Shen, **PRL 108,117403 (2012)**
2. C. Kastl, C. Karnetzky, H. Karl and A.W. Holleitner, **Nat. Comm. 6, 6617 (2015)**
3. D. Hsieh, F. Mahmood, J.W. McIver, D.R. Gardner, Y.S. Lee and N. Gedik, **PRL 107,077401 (2011)**
4. N.P. Butch, K. Kirshenbaum, P. Syers, A.B. Sushkov, G.S. Jenkins, H.D. Drew and J. Paglione, **PRB 81, 241301 (2010)**
5. Y.H. Wang, D.Hsieh, E.J. Sie, H. Steinberg, D.R. Gardner, Y.S. Lee, P.J. Herrero and N. Gedik, **PRL 109, 127401 (2012)**
6. J.S. Barriga, E. Golias, A. Varykhalov, J. Braun, L.V. Yashina, R. Schumann, J. Minár, H. Ebert, O. Kornilov, and O. Rader, **PRB 93, 155426 (2016)**
7. H. Zhang, C.-X. Liu, X.-L. Qi, X. Dai, Z. Fang, and S.-C. Zhang, **Nature Physics 5, 438-442(2009)**
8. J. Zhang, C.-Z. Chang, Z. Zhang, J. Wen, X. Feng, K. Li, M. Liu, K. He, L. Wang, X. Chen, Q. Xue, X. Ma, and Y. Wang, **Nat. Commun 2, 574(2011)**
9. H. Plank, L. E. Golub, S. Bauer, V. V. Bel'kov, T. Herrmann, P. Olbrich, M. Eschbach, L. Plucinski, C. M. Schneider, J. Kampmeier, M. Lanius, G. Mussler, D. Grützmacher, and S. D. Ganichev, **PRB 93, 125434 (2016)**

Summary of revisions

1. We changed the reference for circular photon drag effect to three articles (reference 19, 20, 29):
 19. Shalygin, V. et al. Spin photocurrents and the circular photon drag effect in (110)-grown quantum well structures. JETP letters 84, 570–576 (2007).
 20. Jiang, C. et al. Helicity-dependent photocurrents in graphene layers excited by midinfrared radiation of a CO₂ laser. Physical Review B 84, 125429 (2011).
 29. Belinicher, V. On the mechanisms underlying the circular drag effect. Sov. Phys. Solid State 23, 2012 (1981).
2. We changed the momentum relaxation time in the numerical calculation from 0.1 ps to 1 ps.
3. We added theoretical analysis of the photocurrent generated by the optical transitions between the two Dirac cone surface states in Supplementary Note 5.
4. We added a new section - Note 4 and Figure 5 and 6 in the supplementary material -- to study the variation of the photocurrent with different relaxation times for bulk excited carriers and surface excited carriers. We also added a few sentences in the discussion part of the main text, providing details of the photocurrent contribution from the optical transitions between the two Dirac cones.
5. We changed the notation of the angles including the quarter wave plate rotation angle ($\theta \rightarrow \varphi$) and the angle of incidence ($\varphi \rightarrow \theta$) and the notation of the tensors for CPGE ($\beta \rightarrow \gamma$) and CPDE ($\chi \rightarrow \tilde{T}$).
6. We added a new section – Figure 4 in the supplementary material -- to compare the photocurrent contribution from the valence band to surface state transitions to the contribution from the surface state to conduction band transitions.
7. We added a new figure (Fig 2) in the supplementary material which is the temperature dependence of the longitudinal resistance R_{xx} in Device B. The figure demonstrates that the chemical potential of the sample is located in the bulk band gap.
8. We added a new figure (Fig 8) in the supplementary material which is the photocurrent imaging on Device B with a small beam spot $\sim 5\mu\text{m}$. Correspondingly, a new sentence was added in the discussion part of the main text to refer to the figure and to exclude the contribution to the helicity dependent photocurrent from the laser heating.

9. We added a few sentences in the main text that supports Figure 1. The purpose is to point out the linear polarization dependent photocurrent has been studied by reference 28.
10. We changed the notation for the asymmetric matrix element to $(|M_{\xi\eta}|^2)_a$ for clarity.
11. We added a sentence in the discussion part of the main text to include the difference in the relaxation times for surface and bulk carriers as one possible source for the asymmetry of the helicity dependent photocurrent around the Dirac point.

Reviewers' Comments:

Reviewer #1 (Remarks to the Author):

I have read the revised version of the manuscript by Yu Pan et al. and I believe that the authors replied to my comments in a satisfactory way. I recommend the publication of the manuscript in its current form.

Reviewer #2 (Remarks to the Author):

This revised manuscript is clearly an improvement over the first version. Authors have carefully considered all of referees' comments and addressed those issues in the revised version, or in the rebuttal letter. I am satisfied with author's revision. I think that the manuscript is now good to be recommended for publication in Nature Communication.

Reviewer #3 (Remarks to the Author):

Overall, I think that the authors did a good job at answering all my comments, as well at the comments by the other referees.

Two points which I would like the authors to clarify:

1) In Figure 2 of the supplementary, the minimal temperature is 15K. If this is indeed a bulk insulating sample, with Fermi level in the surface states, R_{xx} should show a saturation at lower temperatures. Is such a saturation seen in these samples?

2) Figure 8 of the supplementary is a great addition and to the point in answering my comment. It does seem, however, that deep in the middle of the sample, there are actually large variations in the photocurrent as a function of space (which seem to be random to the naked eye). What is causing these large variations?

Other than the above comments, I think that the authors have addressed all points adequately. Once these comments are addressed I can recommend that the paper be published in Nature Comm.

We thank all three referees' careful reading and positive opinions about our work. The first two reviewers did not ask new questions, therefore we will only address the questions from the third reviewer.

Reply to the third reviewer:

1) In Figure 2 of the supplementary, the minimal temperature is 15K. If this is indeed a bulk insulating sample, with Fermi level in the surface states, R_{xx} should show a saturation at lower temperatures. Is such a saturation seen in these samples?

Reply: The transport measurements were conducted in an optical continuous flow cryostat where we also carry out the photocurrent measurements; this cryostat does not cool down below 10K. We note that Fig. 2 of the supplementary section does show a trend toward saturation of R_{xx} at the lowest temperatures we studied, though not as cleanly as reported elsewhere (e.g. Tian *et al. Scientific Reports* **4**, 4859 (2014)). Our statements about the chemical potential being in the bulk gap are principally based on the gate voltage dependence shown in Fig. 2 of the manuscript. The CPGE is likely dominated by absorption of light at the top surface directly under the gate. However, we may not be gating both the top and bottom surfaces of the sample equally, resulting in the incomplete saturation of R_{xx} at low temperature.

2) Figure 8 of the supplementary is a great addition and to the point in answering my comment. It does seem, however, that deep in the middle of the sample, there are actually large variations in the photocurrent as a function of space (which seem to be random to the naked eye). What is causing these large variations?

Reply: We do notice some spatial variations in the photocurrent when we used a small beam spot (~5 μm). Such variations have been studied in Bi_2Se_3 by others (e.g. Kastl *et al., Appl. Phys. Lett.* **101**, 251110 (2012)) and attributed to local fluctuations of the potential landscape. Although they reported a domain size of 1-2 μm , we suspect that this is dependent on the substrate and different substrates could lead to a different domain size. Another possible origin of the spatial variation in the photocurrent may arise from the way we extracted the HDPC. In the imaging measurement, we extracted the HDPC by measuring the photocurrent with left-circularly polarized light and right circularly polarized light (independent scanning for each measurement), and then subtracting them. The two scans did not precisely match in space and may vary within 1 μm . Therefore, the extracted HDPC may deviate from the actual value due to the subtraction of photocurrent at different positions. We believe the above two reasons are the major causes of the spatial variation in photocurrent we reported.